# Nasal Chondrosarcoma of the Lower Lateral Cartilage

**DOI:** 10.3390/medicina55050128

**Published:** 2019-05-09

**Authors:** Hannah B. Tan, Joanne Rimmer

**Affiliations:** Department of Otolaryngology, Head and Neck Surgery, Monash Health, Clayton, VIC 3168, Australia; rimmer.joanne@gmail.com

**Keywords:** sinonasal, chondrosarcoma, head and neck sarcoma, lower lateral cartilage, open septorhinoplasty

## Abstract

Head and neck chondrosarcomas are incredibly rare with documented cases arising from skull base, maxilla, larynx, and nasal septum. We present the first reported case of chondrosarcoma arising from the lower lateral cartilage of the nose treated with surgical resection and primary reconstruction.

## 1. Introduction

Sarcomas of the head and neck are rare, comprising less than one per cent of all head and neck neoplasms [1]. Of these few cases, only 20% arise from bone or cartilage origin, most commonly the skull base, maxilla and larynx [1]. There are only a few reported cases of chondrosarcoma arising from nose, most commonly originating from septal cartilage [2,3,4,5]. Nasal tip origin is a rare entity with only two reported cases [6]. We present the first documented case of chondrosarcoma arising from the lower lateral cartilage (LLC) of the nose.

## 2. Case

A 68-year-old man presented with a ten-year history of a lump in the right nasal tip, with a significant increase in size over a six-month period. Medical history included adenocarcinoma of the rectum treated with surgery and radiotherapy two years previously, polycythaemia, chronic alcohol abuse and a 50-pack per year smoking history. Clinical examination revealed a large bulbous firm swelling of the right lower third of the nose causing significant cosmetic asymmetry and distorting the right ala (Figure 1). Rigid nasal endoscopy was otherwise unremarkable, and oropharyngeal and neck examination was normal. Computed tomography (CT) scan showed a 2.8 cm soft tissue lesion arising from the right anterior nares, abutting but not obviously involving the cartilaginous septum. Magnetic resonance imaging (MRI) confirmed a right-sided 3.4 cm mass arising from the lower lateral alar cartilage (Figure 2) with no apparent invasion of the nasal septum or adjacent soft tissues. An incisional biopsy via the vestibular aspect revealed a well-defined avascular mass with a lobulated surface. The initial histopathological report was suggestive of benign enchondroma, however further review considered low-grade chondrosarcoma to be more likely. Imaging of the neck and chest showed no evidence of metastatic disease, but two small lung nodules were identified, with a subsequent diagnosis of primary adenocarcinoma of the lung.

After discussion at the multidisciplinary head and neck meeting, the patient underwent primary resection via an external rhinoplasty approach. The skin and soft tissue envelope (SSTE) was not adherent to the tumor, which was well-circumscribed and easily dissected from the surrounding soft tissue with no evidence of local invasion. Macroscopic resection of a 3 cm × 3 cm × 3 cm mass arising from the lateral crus of the right LLC was performed (Figure 3 and Figure 4). Soft tissue superficial to the tumor and the right upper lateral cartilage were excised as margins. Primary reconstruction of the LLC was performed using native septal cartilage harvested through a separate left Killian’s incision. A lateral crural strut graft was sutured to the preserved dome of the native LLC medially and placed into a soft tissue pocket laterally (Figure 5). Silastic splints were placed on either side of the right ala to minimize the dead space left after resection of the tumour (Figure 6). An external plaster of Paris splint was also applied.

Histopathological review of the specimen confirmed an intermediate grade chondrosarcoma with clear resection margins. As the tumor was resected from an easily observable area, the multidisciplinary head and neck meeting recommendation was for surveillance only, with no postoperative radiotherapy at this stage.

There were no post-operative complications. Sutures and splints were removed one week following surgery. The SSTE was contracted and adhered to underlying tissue without development of seroma or haematoma. At five-week review, the right nasal airway was patent with no dynamic collapse of the right nasal ala. At four months (Figure 7), there was no evidence of locoregional disease and the patient’s right nasal airway remained patent (Figure 8). The patient is due to undergo radiotherapy for the lung cancer diagnosed on his staging CT scan.

## 3. Discussion

As demonstrated in this case, chondrosarcoma is histopathologically difficult to differentiate from benign neoplasms of cartilaginous origin, such as enchondroma, osteochondroma, and synovial chondromatosis [7]. Differential diagnoses to be considered for high-grade chondrosarcoma include chondroblastic osteosarcoma, fibrosarcoma, and malignant fibrous histiocytoma [7].

Surgical resection is the primary modality of treatment for chondrosarcomas [1,5]. A systematic review of 161 patients with sinonasal chondrosarcoma found surgery alone was performed in 72% and a combination of surgery and radiotherapy in 21.7% of cases [5]. Open surgical approach alone was used in 89 patients and endoscopic techniques alone in 10 patients [5]. In this case, the mass was easily accessible through an external rhinoplasty approach. Radiotherapy may be considered in advanced disease, surgically unresectable cases or with positive surgical margins, however, there are no prospective studies to guide which patients would benefit [1,7]. Pre-operative radiotherapy to decrease tumor mass has also been described [5]. Chemotherapy is not supported by current evidence [1,5,7].

Primary reconstruction at the time of resection was both ideal and convenient with native septal cartilage harvested from an adjacent surgical site. The decision was made not to resect the excess SSTE to minimize the risk of wound breakdown, particularly in view of his ongoing smoking history.

As the condition is rare, the existing literature is retrospective in nature, and therefore, data regarding treatment, outcome and prognosis are variable. A wide range of five-year survival rates from 44 to 88% has been reported [5,6,8,9,10,11]. Rate of lymph node (5.6%) and distant metastasis (6.7%) is low [5]. Invasive disease due to local recurrence is reported as the most common cause of death [5]. In a systematic review of 161 patients with sinonasal chondrosarcoma, the recurrence rate in patients receiving surgery alone was 32.8% compared to 29.4% in those receiving combination surgery and radiotherapy [5]. As there are no clinical guidelines regarding adjuvant radiotherapy, there is a significant possibility of selection bias, and therefore, the difference in recurrence rates. Large tumour size, high pathological grade and subtotal resection are characteristics associated with the decision for surgery and radiotherapy compared to surgery alone [5]. However, in the systematic review by Khan et al., tumor size and staging were not adequately reported in the included studies [5]. The average age of patients in the surgery and radiation group was younger compared to surgery alone, but this was skewed by the inclusion of a study of paediatric patients.

A recent retrospective study of 47 patients with grades I and II skull base chondrosarcoma reported local recurrence was 33% in patients receiving surgery alone compared to surgery and adjuvant proton therapy (11%), however, there was no significant difference in 10-year disease-specific survival between the groups [12]. There was no significant difference between the groups regarding patient sex, age, and tumor size. In the surgery and proton therapy group, a greater proportion of cases had petroclival tumors compared to the surgery only group. In addition, the presence of internal carotid artery abutment was significantly greater in the surgery and proton therapy group, which suggests tumor location and proximity to certain anatomical structures affects the decision for adjuvant radiotherapy.

## 4. Conclusions

Although rare, chondrosarcoma must be considered in cases of a lower nasal mass as there is a significant risk of histological misdiagnosis of a benign lesion on biopsy. Surgical access through an external rhinoplasty approach for resection is ideal for both resection and primary reconstruction of tumors located at this site.

## Figures and Tables

**Figure 1 medicina-55-00128-f001:**
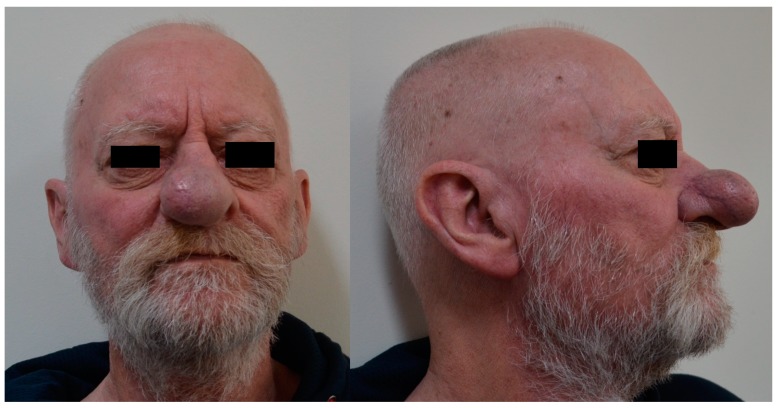
Right nasal tip/vestibule mass.

**Figure 2 medicina-55-00128-f002:**
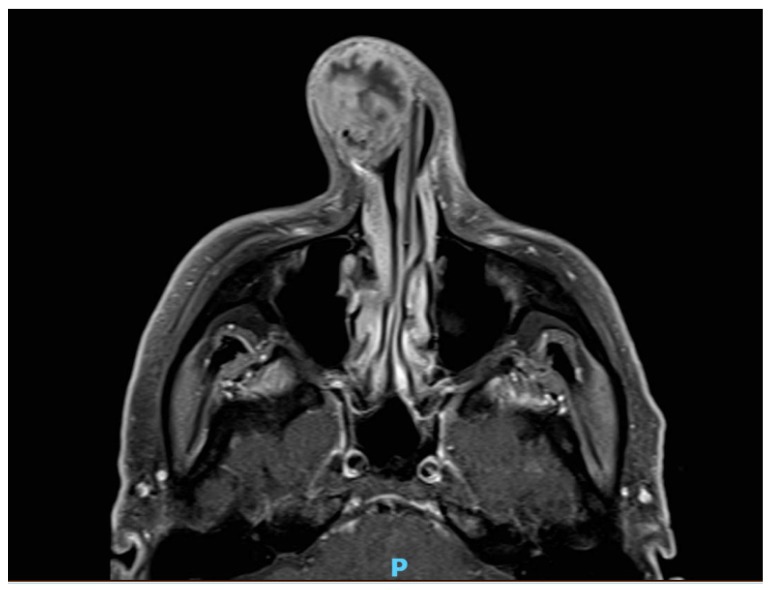
Axial T1 weighted magnetic resonance imaging (MRI) showing mass arising from right lower lateral cartilage.

**Figure 3 medicina-55-00128-f003:**
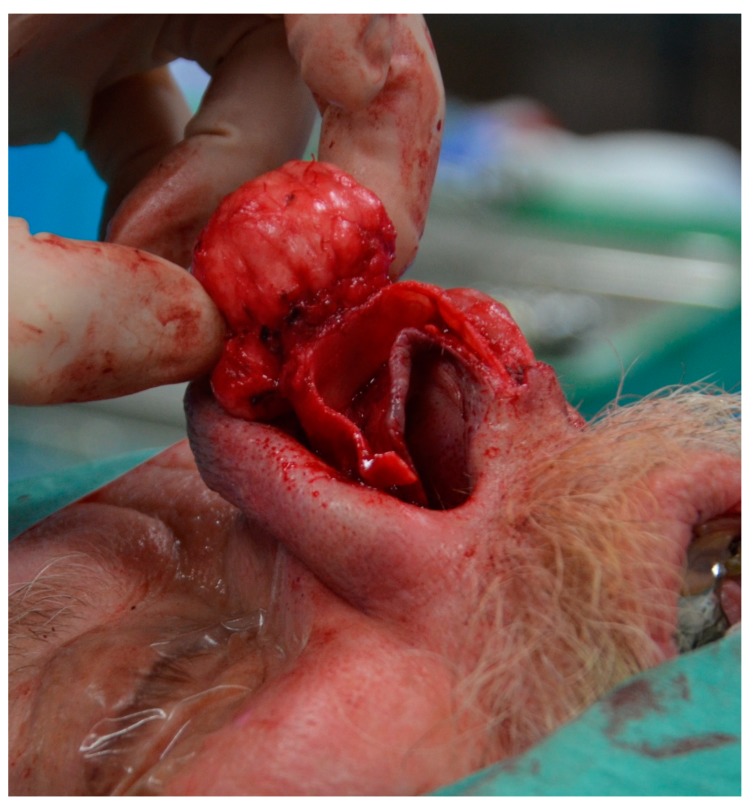
Intraoperative photograph showing mass arising from the right lower lateral cartilage (LLC).

**Figure 4 medicina-55-00128-f004:**
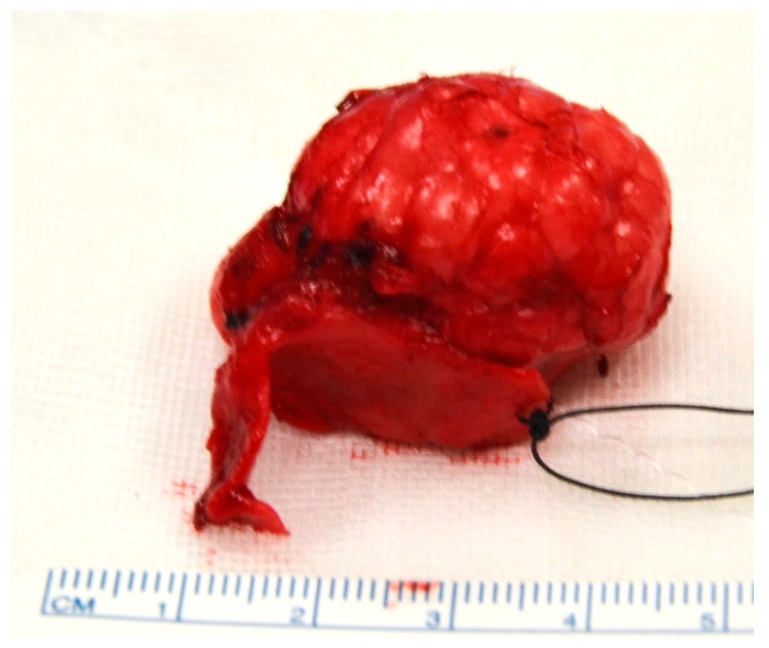
Resected specimen with suture marking the medial resection margin of the LLC.

**Figure 5 medicina-55-00128-f005:**
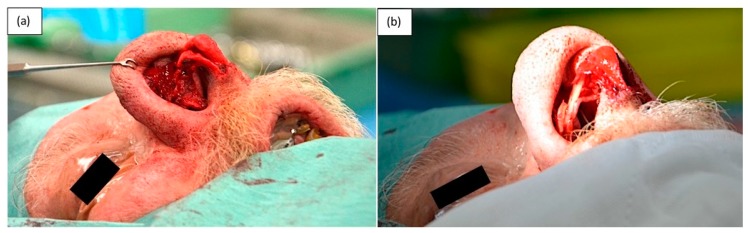
(**a**) Remaining native medial crus and dome of LLC, (**b**) Reconstructed LLC using septal cartilage graft.

**Figure 6 medicina-55-00128-f006:**
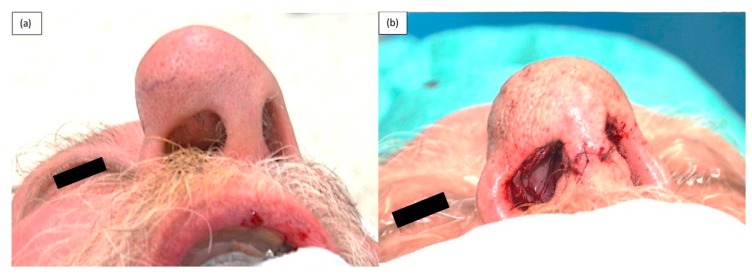
(**a**) Pre-operative photograph showing expansion of the skin and soft tissue envelope (SSTE) due to mass, (**b**) Immediate post-operative photograph demonstrating residual expansion of SSTE.

**Figure 7 medicina-55-00128-f007:**
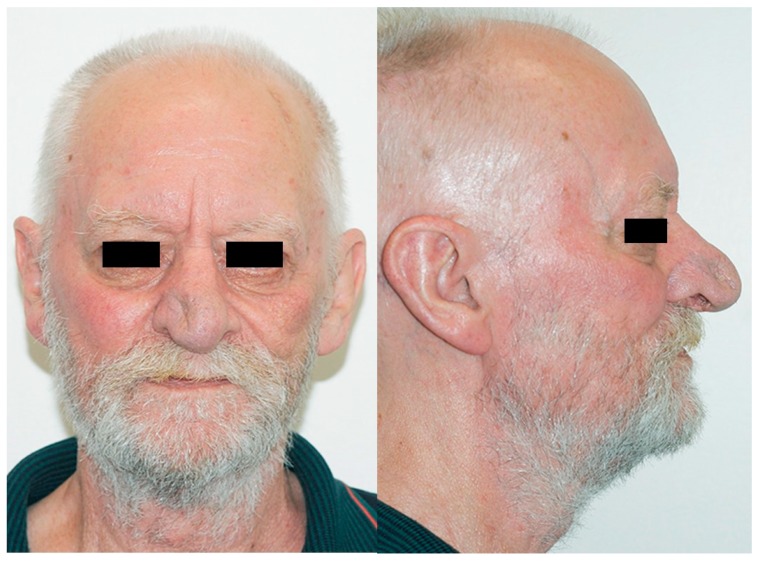
Four-months post-operative.

**Figure 8 medicina-55-00128-f008:**
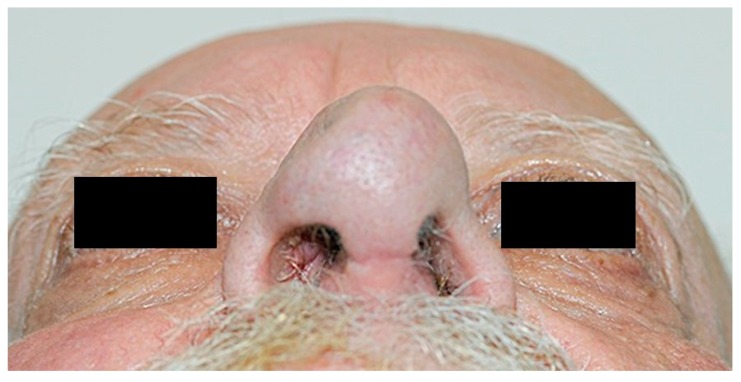
Four-months post-operative photograph showing adequate support of right ala.

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
