# Peer review of "Nasal Chondrosarcoma of the Lower Lateral Cartilage"

_medicina, 2019, doi:10.3390/medicina55050128_

Round 1

Reviewer 1 Report

The studies cited comparing surgical versus surgical plus radiotherapy should be presented relative to whether the tumors in the 2 groups were similar (e.g. local invasion, location) if this data is available.

Author Response

Thank you to the reviewer for their suggestion. We have amended the manuscript to address this comment. 

Reviewer 2 Report

Thanks for the opportunity to read your case report.

While the case would appear to be unique (in the literature at least) it does not in my opinion add particularly to what is known on the topic i.e. rare tumour, arising from cartilage, can be difficult to accurately diagnose, merits surgical treatment etc. The case report is well written and nicely presented. I will leave it to the discretion of the Editor as to whether such a case report reaches the threshold for publication in this journal.

Author Response

We thank the reviewer for their comments. 

Reviewer 3 Report

Well written case report of significant interest. 

Author Response

(The authors gave the same response as above.)
